

# Understanding Aerosol-Cloud Interactions in a Single-Column Model: Intercomparison with Process-Level Models and Evaluation against ACTIVATE Field Measurements

Shuaiqi Tang[1], Hailong Wang[1], Xiang-Yu Li[1], Jingyi Chen[1], Armin Sorooshian[2,3], Xubin Zeng[2], Ewan Crosbie[4,5], Kenneth L. Thornhill[4], Luke D. Ziemba[4], Christiane Voigt[6,7]

[1]Atmospheric, Climate, and Earth Science Division, Pacific Northwest National Laboratory, Richland, WA, USA
[2]Department of Hydrology and Atmospheric Sciences, The University of Arizona, Tucson, AZ, USA
[3]Department of Chemical and Environmental Engineering, The University of Arizona, Tucson, AZ, USA
[4]NASA Langley Research Center, Hampton, VA, USA
[5]Analytical Mechanics Associates Inc., Hampton, VA, USA
[6]Deutsches Zentrum für Luft- und Raumfahrt, Oberpfaffenhofen, Germany
[7]Johannes Gutenberg-Universität, Mainz, Germany

*Correspondence to*: Shuaiqi Tang (shuaiqi.tang@pnnl.gov) and Hailong Wang (hailong.wang@pnnl.gov)

**Abstract.** Marine boundary-layer clouds play a critical role in the Earth's energy balance. Their microphysical and radiative properties are highly impacted by ambient aerosols and dynamical forcings. In this study, we evaluate the representation of these clouds and related aerosol-cloud interactions processes in the single-column version of E3SM climate model (SCM), against field measurements collected during the NASA ACTIVATE campaign over the western North Atlantic, as well as intercompare with high-resolution process-level models. Results show that E3SM-SCM, driven by the ERA5 reanalysis, reproduces the cloud properties as good as the high-resolution WRF simulations. For stronger surface forcings combined with a weaker subsidence taken from a WRF cloud-resolving simulation, both E3SM-SCM and WRF large-eddy simulation produce thicker clouds. This indicates that a proper combination of large-scale dynamics, sub-grid scale parameterizations, and model configurations is needed to obtain optimal performance of cloud simulations. In the E3SM-SCM sensitivity tests with fixed dynamics but perturbed aerosol properties, higher aerosol number concentration leads to more numerous but smaller cloud droplets, resulting in a stronger shortwave cloud forcing (i.e., stronger radiative cooling). This apparent Twomey effect is consistent with prior climate model studies. Cloud liquid water path shows a weakly positive relation with cloud droplet number concentration associated with precipitation suppression, which is different from the nonlinear relation approximated from prior observations and E3SM studies, warranting future investigation. Our findings indicate that the SCM framework is a key tool to bridge the gap between climate models, high-resolution models, and field observations to facilitate process-level understanding.



## 1 Introduction

Marine boundary layer (MBL) clouds are the dominant cloud type over oceans, with an annual mean occurrence frequency of 45% (Warren et al., 1988) and coverage of 34% including stratocumulus, stratus and fog (Warren et al., 1988) or 23% for stratocumulus only (Wood, 2012). Its high reflectivity overlapped with low-reflective ocean surface underneath leads to a strong shortwave cooling effect, but its longwave warming effect is neglectable due to low cloud top height (Hartmann et al., 1992). In global climate models (GCM), the representation of MBL clouds and their radiative effects has long been a challenging task (e.g., Bony and Dufresne, 2005; Brunke et al., 2019). Even the latest Coupled Model Intercomparison Project Phase 6 (CMIP6) models still have a large inter-model spread in the cloud shortwave effect (Bock et al., 2020) that introduces large uncertainties to climate projection.

The western North Atlantic Ocean (WNAO) is one of the regions dominated by MBL clouds. The Gulf Stream with large spatial gradient in sea surface temperature (SST), strong synoptical systems such as tropical and extratropical cyclones, aerosols generated locally or transported from the adjacent North American continent, all contribute to the complex aerosol-cloud-meteorology-ocean interactions over this region (e.g., Painemal et al., 2021; Corral et al., 2021). Recently, Sorooshian et al. (2020) provided an overview of the past atmospheric studies over WNAO region, followed by more detailed overviews on circulation, boundary layer and clouds (Painemal et al., 2021), clouds and precipitation (Kirschler et al., 2023), and atmospheric chemistry and aerosols (Corral et al., 2021). However, among 715 peer-reviewed publications between 1946 and 2019, only 2% of the studies are related to aerosol-cloud interactions (ACI) (Sorooshian et al., 2020). This indicates that ACI over WNAO region is underexplored, considering that ACI has long been emphasized as the largest uncertainty source in climate model simulations (IPCC, 2013, 2021).

With the limited prior understanding, a three-year field campaign, Aerosol Cloud meTeorology Interactions oVer the western ATlantic Experiment (ACTIVATE) project (Sorooshian et al., 2019), was conducted between 2020 and 2022 targeting the complex ACI for MBL clouds over the WNAO region. Two aircraft flew simultaneously in spatial coordination: a low-flying aircraft making in-situ measurements and a high-flying aircraft making remote-sensing measurements and releasing dropsondes. Among the total of 162 flights, 12 of them were conducted as "process study" flights (Sorooshian et al., 2023), during which the flying patterns of the two flights were carefully designed to provide detailed information about the scene encompassing the clouds of interest. In some cases, including the case chosen for this study, the high-flying aircraft released numerous dropsondes along a large circle and the low-flying aircraft conducted stacked below-, in-, and above-cloud flight legs within the circle. The dropsonde-derived divergence profiles and surface fluxes have been used to constrain process-level modelling studies (Chen et al., 2022; Li et al., 2022; Li et al., 2023).



A few process-level studies have been conducted using the Weather Research and Forecasting (WRF) model nested domain
regional simulation (Chen et al., 2022) and WRF large-eddy simulation (LES) (Li et al., 2022; Li et al., 2023). The WRF
regional simulation has an inner domain at 1 km convection-permitting horizontal grid spacing, hereafter referred to as
cloud-resolving model (CRM) simulation in this study. Note that this is different from the conventionally defined CRM,
which is usually run with prescribed large-scale forcing, periodic boundary conditions, in a limited region analogous to a
single-column model (SCM) (Randall et al., 1996). A post-frontal MBL cloud case related to a winter cold-air outbreak
(CAO) was studied in these CRM and LES studies. Chen et al. (2022) successfully simulated the observed cloud roll
structure in WRF-CRM. They found that a distinctive boundary layer wind direction shear favours the formation and
persistence of cloud rolls. Li et al. (2022) validated the ERA5-derived large-scale forcing with dropsonde-derived forcing
and tested the sensitivity of WRF-LES to the large-scale forcing. They furthermore investigated ACI with a series of LES
sensitivity experiments based on spatial variability in aircraft-measured aerosol and cloud properties (Li et al., 2023).

In this study, we focus on SCM simulations for the same CAO case investigated in Chen et al. (2022); Li et al. (2022); Li et
al. (2023). As these models simulate the same case in different complexity and resolution, we are now able to make detailed
process-level analysis of ACI through the multi-scale LES-CRM-SCM intercomparison. This is a step further than studies
using individual model. Our first goal is to understand how the CAO-related post-frontal MBL clouds are simulated in the
SCM in contrast to the LES and CRM simulations, and the observations. Another goal is to explore how the simulated MBL
clouds respond to perturbations of aerosol properties prescribed into the SCM through sensitivity studies using observations
collected during the ACTIVATE campaign. We introduce the selected case, data, and models in Sect. 2, intercompare SCM
with CRM and LES results in Sect. 3, and then show results of SCM sensitivity studies in Sect. 4. Conclusion remarks are
provided in Sect. 5.
**2 Case Description, Observations and Simulations**
**2.1 The CAO case on 1 March 2020**
This study focuses on a CAO case observed on 1 March 2020, after the passage of a cold front. A large area of MBL clouds
formed associated with warm SST, cold air advection, and large-scale subsidence. The ACTIVATE campaign deployed two
spatially coordinated aircraft to measure the post-frontal MBL clouds from different heights (Fig. 1a). The High Spectral
Resolution Lidar – generation 2 (HSRL-2) from the high-flying King Air aircraft measured vertical aerosol backscattering
profiles, which were used to estimate the cloud top height. The King Air also released 11 dropsondes in a ~110 km diameter
circle centered near (38.1˚N, 71.7˚W) to measure the vertical profiles of the meteorology state. The low-flying Falcon
aircraft mainly provided in-situ aerosol and cloud microphysical measurements. The entire Falcon flight is divided into many
flight "legs" (Dadashazar et al., 2022b). Each flight leg represents a segment during which the flight is measuring under a
specific condition at constant altitude (e.g., below/in/above cloud) or is in a specific operation mode (e.g., ascending,



descending). For most of this study, we focus on eight flight legs within or near the dropsonde array domain (Fig. 1b),
including two minimum-altitude (MinAlt) legs, two below-cloud-base (BCB) legs, one above-cloud-base (ACB) leg, two
below-cloud-top (BCT) legs, and one above-cloud-top (ACT) leg. The first six flight legs were stacked in different heights as
a "wall" pattern. The last two legs were flying outside the dropsonde domain but used here for sensitivity study purpose.

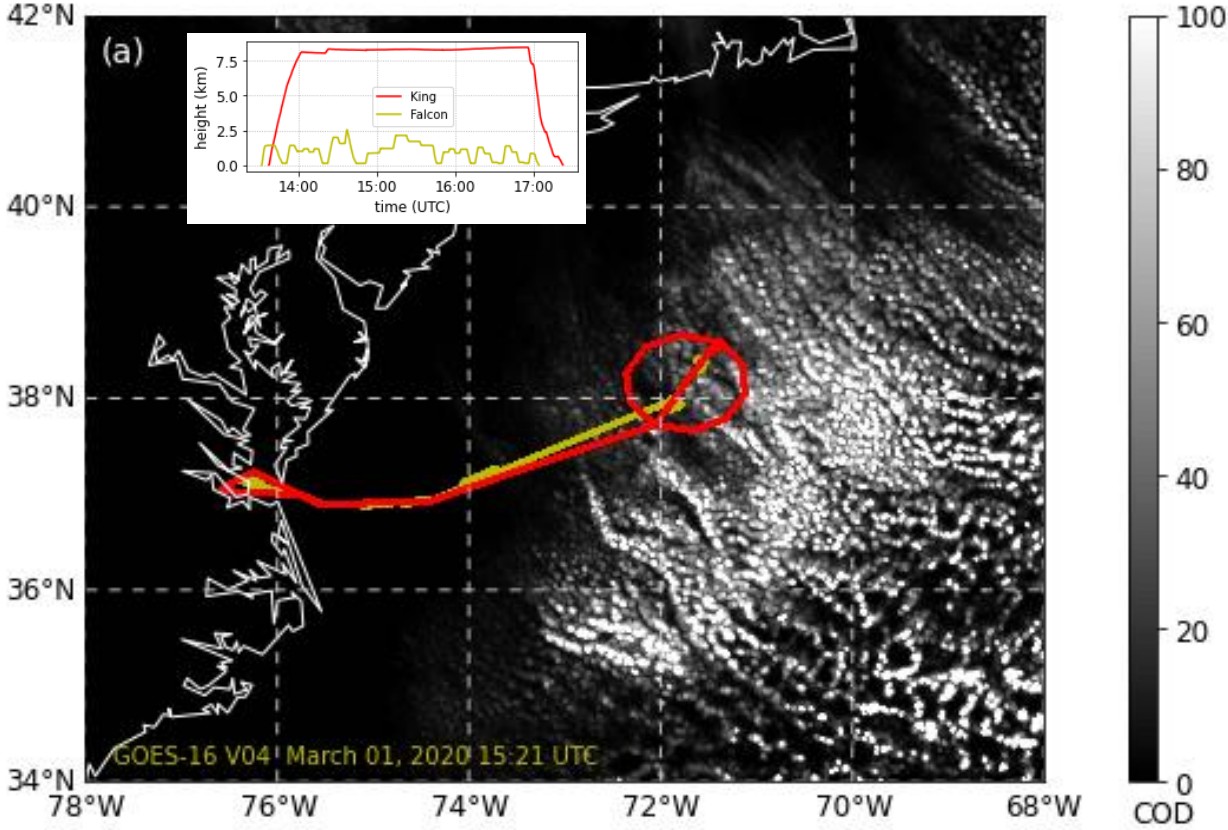

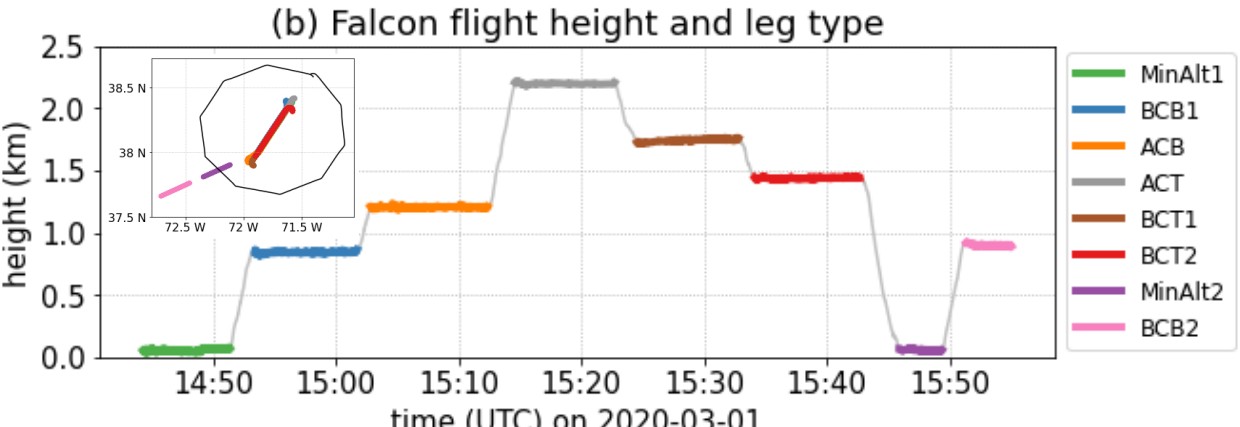




**Figure 1: (a). ACTIVATE flight tracks for Falcon (yellow) and King Air (red) aircraft on 1 March 2020 (RF13), overlaid with GOES-16 satellite-measured cloud optical depth (COD) at 15:21 UTC. The insert shows the time series of flight altitude for both aircraft. (b) Time and height of the eight Falcon flight legs within or near the dropsonde array domain. The insert is the horizontal location of the eight flight legs and the dropsonde domain (thin black line). Acronym of flight leg types: BCB: below cloud base; ACB: above cloud base; ACT: above cloud top; BCT: below cloud top; MinAlt: minimum altitude (~120 m above ground level (AGL)).**

**2.2 Forcing and Evaluation Data**

Table 1 lists the aircraft measurements used in this study. These observational data are used mainly for two purposes: driving models as initial and boundary conditions and evaluating model results. Satellite measurements and reanalysis data are also used to supplement the aircraft measurements to give a more complete view and fill data gaps when aircraft data are unavailable. Specifically, the liquid water path (LWP) and the ice water path (IWP) are retrieved from GOES-16 geostationary satellite using the Visible Infrared Solar-Infrared Split Window Technique (VISST) (Minnis et al., 2008; Minnis et al., 2011) algorithm from the NASA-Langley Satellite Cloud Observations and Radiative Property retrieval System (SatCORPS). ERA5 reanalysis (Hersbach et al., 2020) is used to provide model initial and boundary conditions to drive the WRF-CRM simulation, and to supplement the large-scale forcing used by WRF-LES and E3SM-SCM. More details of the large-scale forcing are given in the next subsection.

**Table 1: Aircraft measurements used in this study.**

| Instrument | Measurements | Platform | Data Version |
|---|---|---|---|
| GPS | Flight location (lat, lon, alt) | Falcon | R4 |
| N/A | Flight leg flag | Falcon | R3 |
| Five-port pressure system (TAMMS) | 3-D winds | Falcon | R4 |
| Rosemount 102 sensor | Temperature | Falcon | R4 |
| Diode laser hygrometer (DLH) | Water vapor mixing ratio | Falcon | R1 |
| Scanning Mobility Particle Sizer (SMPS) | Aerosol number size distribution (2.97 – 94.0 nm) | Falcon | R4 |
| Laser Aerosol Spectrometer (LAS) | Aerosol number size distribution (93.9 – 3487.5 nm) | Falcon | R3 |
| High-Resolution Time-of-Flight Aerosol Mass Spectrometer (AMS) | Mass concentration of aerosol composition (Organic, Sulphate, Nitrate, Ammonium, Chloride) | Falcon | R2 |
| Cloud Condensation Nuclei (CCN) Counter | CCN number concentration with supersaturation (SS) scanning from ~ 0.16% to 0.72% | Falcon | R0 |
| Fast Cloud Droplet Probe (FCDP) | Cloud droplet number size distribution (3 – 50 µm), liquid water content (LWC), droplet number concentration and effective radius | Falcon | R1 |
| GPS | Flight location (lat, lon, alt) | King Air | R0 |
| High Spectral Resolution Lidar (HSRL-2) | Cloud top height | King Air | R0 |
| Dropsonde (Vömel et al., 2023) | Temperature, pressure, altitude, relative humidity, U | King Air | R1 |





| | wind, V wind | | |
|---|---|---|---|

**2.3 Model Simulations**

The SCM used in this study is based on the Energy Exascale Earth System Model (E3SM) version 2 (Golaz et al., 2022; Bogenschutz et al., 2020). It includes a deep convective parameterization from Zhang and McFarlane (1995) with modification from Xie et al. (2019) to improve diurnal cycle of precipitation, a two-moment microphysics from Gettelman and Morrison (2015) (MG2), and a Cloud Layers Unified By Binormals (CLUBB) (Golaz et al., 2002; Larson and Golaz, 2005) parameterization for turbulence, shallow convection and macrophysics all-together. Some parameters of these schemes were systematically re-tuned to improve the overall performance of subtropical stratocumulus clouds (Ma et al., 2022). Aerosols generally require long spin-up time that is unrealistic during the relatively short SCM case durations. Instead of directly use the aerosol scheme, three options has been implemented in E3SM-SCM to treat aerosols: specifying droplet and ice number concentrations to "bypass" ACI, using "prescribed" aerosols from a 10-year E3SM climatology simulation under present-day forcing conditions, or using "observed" aerosol information if available (Bogenschutz et al., 2020). The information of three lognormal distribution modes of aerosols (Aitken, accumulation and coarse) is needed in the "prescribed" and "observed" methods to replace the output from the aerosol scheme, which is 3-mode Modal Aerosol Module (MAM3) (Liu et al., 2012) in the E3SM SCM configuration. Note that this differs from the default MAM4 scheme (Liu et al., 2016) in E3SM GCM. The "observed" method currently does not include vertical variation of aerosols (i.e., observed aerosol information is applied to all vertical layers from the surface to model top). Therefore, to investigate ACI and the impact of aerosol vertical distribution on clouds, we use a "prescribed-observed" hybrid method in this study, in which we replace the prescribed aerosol data with aircraft-measured aerosols or idealized conditions.

E3SM-SCM is driven by prescribed large-scale forcing data (i.e., advective tendencies and vertical velocity) and surface turbulent fluxes, with a nudging timescale of 3 h to reduce biases in the atmospheric mean state. We use the same forcing data as Li et al. (2022) in their WRF-LES simulations over the dropsonde region (red circle in Fig. 1a). The large-scale forcing fields are shown in the left panel of Fig. 2. The environment exhibits strong subsidence with cold and dry advection in the lower atmosphere. The near-surface cold and dry air and relatively high SST (not shown) lead to large surface latent (~ 400 W/m$^2$) and sensible (> 200 W/m$^2$) heat fluxes. Although these data are obtained from the ERA5 reanalysis, which exhibits a cold and dry bias in MBL (Seethala et al., 2021), the wind structure is well captured (Chen et al., 2022) and the ERA5 divergence agrees well with that derived from the ACTIVATE dropsonde array (Li et al., 2022). Overall, it has been shown that the ERA5-derived large-scale forcing and surface turbulent fluxes can reasonably reproduce clouds and boundary layer for this case in WRF-LES simulations (Li et al., 2022; Li et al., 2023).

The WRF-CRM (Chen et al., 2022) and WRF-LES (Li et al., 2022; Li et al., 2023) simulations are also used for intercomparison with the E3SM-SCM. The WRF-CRM has an outer domain at 3 km horizontal grid and an inner domain in



1 km convective-resolving resolution, with an interactive land option and prescribed SST from ERA5. It is able to reproduce
the "cloud street" feature seen in satellite images (Chen et al., 2022). Over the dropsonde region, the nested WRF-CRM
simulation shows stronger cold advection in MBL and weaker subsidence above MBL (the right panel of Fig. 2) than the
ERA5 large-scale forcing. The near-surface temperature and moisture in WRF-CRM are lower than ERA5, yielding higher
surface latent (21–68 W/m$^2$ higher) and sensible (26–55 W/m$^2$ higher) heat fluxes. The WRF-LES simulation has a domain
size of 60x60 km$^2$ with a 300 m horizontal grid spacing (Li et al., 2022). Its large-scale forcing and surface turbulent fluxes
are prescribed from ERA5, as described above. Nudging is applied only to horizontal winds at a timescale of 1 h, with
temperature and moisture freely evolving. In both CRM and LES simulations, a uniform cloud droplet number concentration
($N_d$) was specified so ACI processes are bypassed. The specified $N_d$ value of 450 cm$^{-3}$ was obtained from a previous version
of FCDP measurements (Li et al., 2022). The newer version of FCDP (see Table 1) with an update instrument calibration
gives a smaller $N_d$ value. As will be seen later (e.g., Fig. 5), E3SM-SCM simulation is more consistent with the updated
FCDP data. Note that here we keep the original setups of prescribed $N_d$ in CRM and LES for consistency with previous
studies (Chen et al., 2022; Li et al., 2022; Li et al., 2023). As all the simulations are available for the same case, we have the
opportunity to demonstrate the value of combining CRM and LES with SCM for process-level understanding of ACI.





**Figure 2: Large-scale environmental conditions, large-scale forcing (horizontal advection and vertical velocity), and surface forcings (latent and sensible heat fluxes) over the dropsonde region from ERA5 used in SCM and WRF-LES (left) and from the WRF-CRM simulations (right). The black lines in large-scale forcing panels mark the zero contour.**

## 3 SCM/CRM/LES intercomparison

All the E3SM-SCM, WRF-LES, and WRF-CRM simulations are initiated at 06:00 UTC, 1 March 2020. With a quick initial spin-up, marine CAO clouds develop between 1 and 2 km above ground level (AGL), then display a gradual reduction in vertical extent, cloud top height, and cloud water content (Figs. 3 and 4). Both SCM and WRF-LES generate a 100% cloud fraction most of the time, while the WRF-CRM simulated cloud fraction decreases with time. This is associated with the success of capturing cloud roll structure in WRF-CRM (Chen et al., 2022). However, this roll structure fails to be simulated in WRF-LES and is neither resolved nor parameterized at the sub-grid scale in E3SM-SCM. Both liquid and ice





hydrometeors are produced and transformed into rain and snow particles. The total ice (including snow) water content is
about one order of magnitude smaller than total liquid water (including rain) (Fig. 3). In our further analyses, we ignore ice
and only focus on liquid clouds for simplicity. All simulations produce a weak mean surface precipitation of less than 2
mm/day (Fig. 4b), except an LES sensitivity experiment discussed later. The evaluation of surface precipitation versus
observations is not conducted here due to the lack of surface measurements and the limited ability of satellite measurements
in detecting weak precipitation from low-level MBL clouds (e.g., Battaglia et al., 2020).

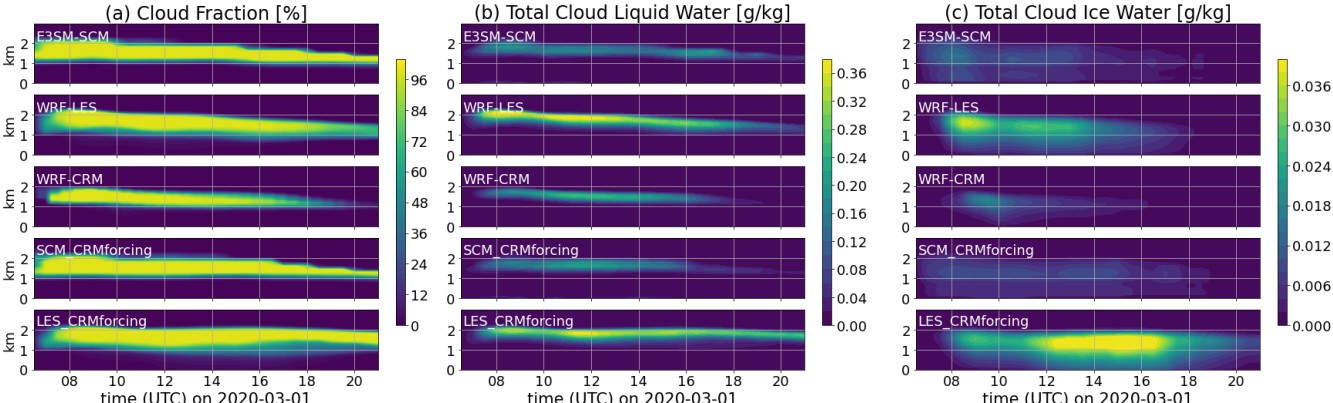

**Figure 3: Time-height cross-sections of cloud fraction, total liquid water, and total ice water produced from different model**
**simulations.**
Figure 4a shows the time series of cloud top height compared with HSRL-2 measurements from the King Air aircraft. The
cloud top heights in models are derived by integrating cloud-fraction-weighted height levels downward, as described in
Varble et al. (2023). E3SM-SCM and WRF-LES produce similar cloud top heights (Fig. 4a), consistent with the highest
observed cloud tops but a few hundred meters higher than most of the aircraft in-situ observations during the time of
operation. It should be noted that HSRL-2 detects the top of each individual cloud, which is usually lower than or, at best,
equal to the highest cloud top within the area. Therefore, this result indicates that cloud top height is reasonably simulated in
the three models, although the HSRL-2 measurements indicate a strong spatial variability. Ignoring the model spin-up period
and high solar zenith angle when satellite retrievals encounter large biases, E3SM-SCM and WRF-CRM also reproduced
total liquid path, while WRF-LES overestimates it by ~50% after 14:00 UTC, compared to the satellite retrievals (Fig. 4c).
For the total ice water (including snow), with only a few valid data points in GOES-16 retrievals, SCM and LES seem to
overestimate it, albeit the overall magnitude is small (Fig. 4d).



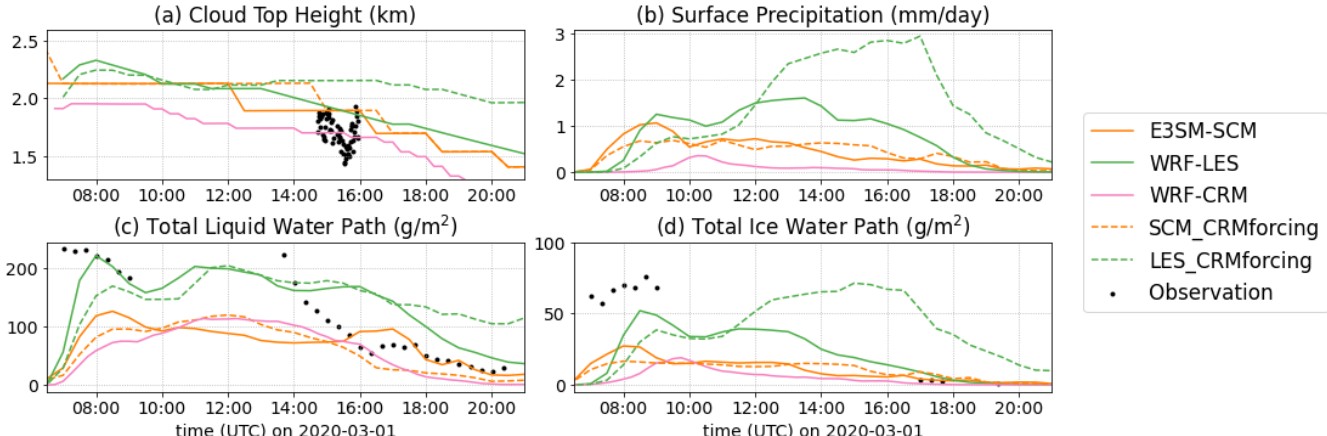

**Figure 4: Time series of model simulations (lines) compared with observation (dots) for the 01 March 2020 case. Observational data are from the King Air HSRL-2 for cloud top height and GOES-16 retrievals for total liquid (including rain) and total ice (including snow) water paths, for which data points at solar zenith angle greater than 65° are removed.**

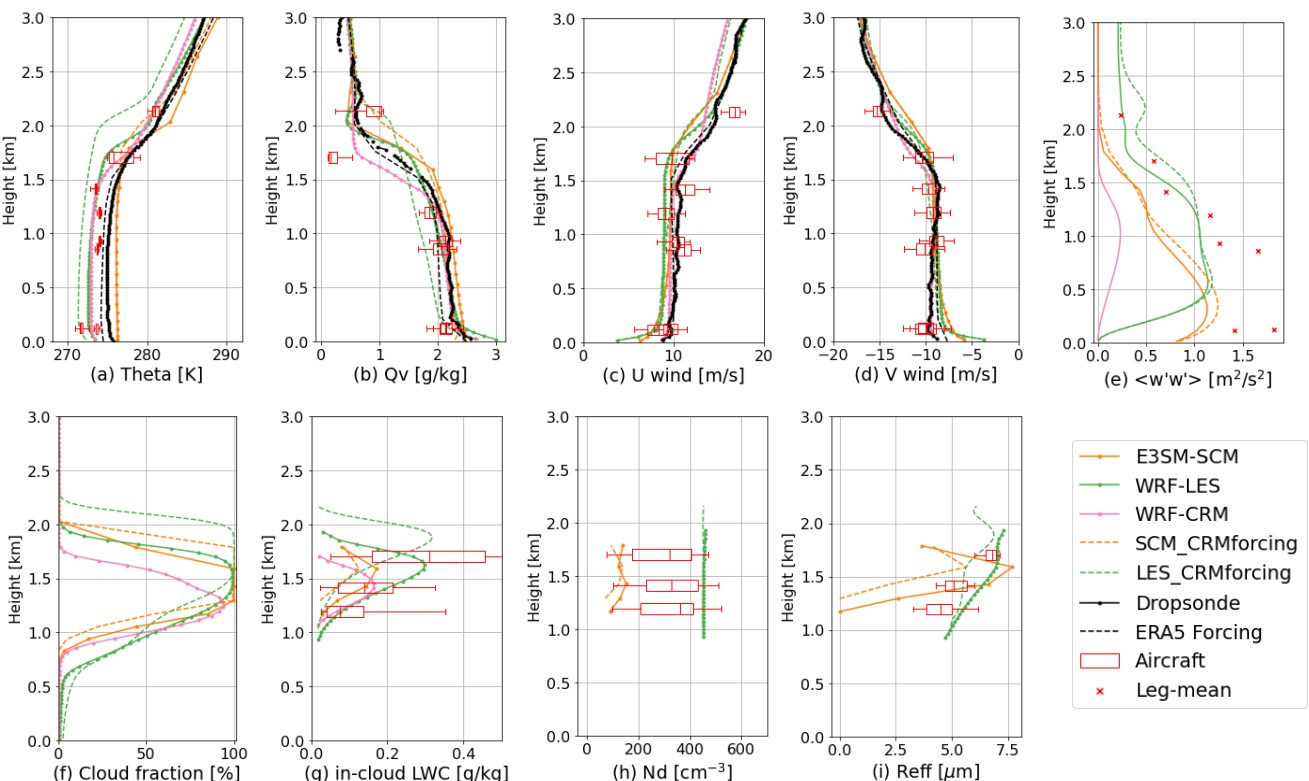

**Figure 5: Vertical profiles of atmospheric state, vertical velocity variance and cloud variables over the analysis domain compared with dropsonde and Falcon measurements. Model profiles are averaged between 15:00 and 16:00 UTC during the aircraft measurements. The box plots indicate the interquartile ranges of the aircraft measurements in each flight leg and the whiskers indicate 5th and 95th percentiles, while the red crosses represent vertical velocity variances calculated from 1 Hz measurements in each flight leg. For cloud microphysical variables, a threshold of in-cloud liquid water content of 0.02 g/m³ and cloud droplet number of 20 cm⁻³ is applied for both model results and aircraft measurements.**



Figure 5 shows the vertical profiles of atmospheric state and cloud variables compared to dropsondes, ERA5 forcing data,
and in-situ aircraft measurements. The atmospheric state variables are constrained by ERA5 reanalysis, which has a colder
and dryer boundary layer than the dropsonde measurements (Figs. 5a and 5b, as well as reported in Seethala et al., 2021).
However, the Falcon data in the boundary layer are also colder and dryer than the dropsonde measurements. These
differences reflect observational uncertainties to some extent. All models are generally consistent with the observations.
However, they do show different temperature biases: E3SM-SCM tends to be warmer while WRF-LES and WRF-CRM tend
to be colder than the dropsondes. This bias is seen throughout the entire simulation period (not shown), indicating different
performances of model parameterizations in E3SM-SCM and WRF-LES, as they used the same initial conditions and large-
scale forcing.

WRF-LES and WRF-CRM both use prescribed $N_d$ obtained from a previous version of Falcon aircraft measurements during
the ACB flight leg, which is higher than the re-calibrated value in the current version (Fig. 5h). They produce similar in-
cloud liquid water content (LWC) below 1.5 km, but WRF-CRM produces lower LWC above 1.5 km because of its lower
cloud top height (Fig. 5g). WRF-LES produces slightly greater droplet effective radius ($R_{eff}$) than aircraft measurements
(Fig. 5i). Together with the large $N_d$, both contribute to large cloud LWC and LWP. WRF-CRM uses bulk microphysics and
does not have $R_{eff}$. The E3SM-SCM simulated LWC is consistent with aircraft measurements during the BCT2 flight leg
near 1.4 km AGL, but lower than the other two in-cloud flight legs (Fig 5g). It also produces larger sizes of cloud droplets
around 1.5 km AGL (Fig. 5i), but produces much lower $N_d$ (Fig 5h). The lower $N_d$ is partly due to the smaller vertical
velocity variance in the SCM simulations compared to the aircraft measurements (Fig. 5e), suggestive of weaker updraft
velocity causing lower supersaturation (SS) which activates fewer cloud condensation nuclei (CCN) into cloud droplets (e.g.,
Kirschler et al., 2022). Another reason is the use of climatological aerosols as input, which provides too low CCN
concentrations for this case. As will be seen in Sect. 4.1, using observed aerosols brings $N_d$ much closer to the observations.

The differences in large-scale forcing and surface turbulent fluxes between ERA5 and WRF-CRM (Fig. 2) raise a question
of how the large-scale forcing impacts the simulations in E3SM-SCM and WRF-LES, considering that WRF-CRM and
E3SM-SCM/WRF-LES show many similarities in simulated cloud properties. To answer this, we configure E3SM-SCM and
WRF-LES with the large-scale forcing and surface fluxes from WRF-CRM over the dropsonde domain (shown in the right
panel of Fig. 2) to conduct two simulations, referred to as SCM_CRMforcing and LES_CRMforcing, respectively. Results of
these two simulations are included as dashed lines in Figs. 3-5. Because of the stronger cold and dry air advection and
weaker subsidence, both SCM_CRMforcing and LES_CRMforcing simulations generate a colder, dryer, and deeper
boundary layer (Figs. 5a and 5b), especially for LES_CRMforcing in which temperature and moisture are not nudged. The
cloud layers in both models are overall thicker than using the ERA5 forcing (Fig. 3a), but detailed features are different
between SCM and LES. Compared to the E3SM-SCM, SCM_CRMforcing follows the same trend of cloud top reduction
rate (Fig. 4a), with a little time lag. Therefore, the cloud grows higher between 15:00 and 16:00 UTC (Fig. 5f) but has





smaller LWC and $R_{eff}$ (Figs. 5g and 5i). For LES, the cloud top height in LES_CRMforcing reduces with a slower rate (Fig.
4a), causing a ~500 m higher cloud top between 15:00 and 16:00 UTC (Fig. 5f). Because of the colder temperature, more
cloud hydrometeors are converted to the ice phase (Figs. 3c and 4d), with more precipitation falling to the ground (Figs. 4b).
This sensitivity study shows a large impact of the large-scale forcing and surface fluxes on cloud properties in the SCM and
LES simulations. A proper combination of large-scale dynamics, sub-grid scale parameterizations, and model configurations
is needed to obtain optimal performance in simulating MBL clouds.

**4 SCM Sensitivity Tests**

The previous section suggests that the underestimation of $N_d$ in E3SM may be due to the underestimation of aerosol number
concentration in the climatological aerosol input for this CAO case. In this section, we use observed aerosols to drive E3SM-
SCM and conduct two sets of sensitivity studies on aerosol number size distribution and composition to investigate how the
input aerosol properties impact clouds and the radiative forcings.

**4.1 Sensitivity to different aerosol number size distributions**

We firstly test the sensitivity of SCM simulations to different aerosol number size distributions using the measurements from
five out-of-cloud legs within or near the dropsonde domain (Fig. 1b). The Falcon aircraft during ACTIVATE campaign was
equipped with an SMPS and an LAS (Table 1) to measure aerosol number size distribution from 2.97 to 94.0 nm (for SMPS)
and 93.9 to 3487.5 nm (for LAS), respectively. We merge the two instruments and fit them into three lognormal modes:
Aitken, accumulation, and coarse modes. For the three parameters in the lognormal distribution function: mode total number
concentration (N), mode geometric median diameter (μ) and standard deviation ($\sigma_g$), we only fit N and μ. Because $\sigma_g$ is also
prescribed in other parts of the model (e.g., radiation calculation), we fix $\sigma_g$ with the E3SM-prescribed values (1.6 for
Aitken, 1.8 for accumulation and coarse) for consistency. A sensitivity test shows that using freely fitted N, μ, and $\sigma_g$ in
E3SM-SCM only yields a minor difference compared to using fixed $\sigma_g$ (not shown). For most flight legs, the fitting of
coarse mode aerosols encountered large uncertainties due to too few samples and large variations. As the coarse mode
aerosol number concentration is usually orders of magnitude smaller than that of the Aitken and accumulation modes, the
poor fitting of coarse mode aerosols is not expected to impact the cloud microphysical properties much.

The centre panel of Fig. 6 shows the fitted aerosol number size distributions from different flight legs, overlapped with
E3SM climatological aerosols near the cloud base height (~900 m AGL). The individual fitting of the three modes as well as
the fitting parameters in each flight leg are shown in the surrounding panels. It is clearly seen that the below-cloud flight legs
(minAlt and BCB) generally have more aerosols, especially in the accumulation mode, than the above-cloud-top flight leg
(ACT). The E3SM climatological aerosols at the BCB2 level show more and larger Aitken mode particles and less coarse
mode particles than all flight leg measurements. For accumulation mode particles that are most important for CCN number



concentration, the E3SM climatology lies between the ACT leg and below-cloud legs. Although the ACT leg does not

represent cloud-base aerosol conditions that are more relevant to the aerosol activation process, the inclusion of this leg

provides information of how SCM performs in a clean environment.

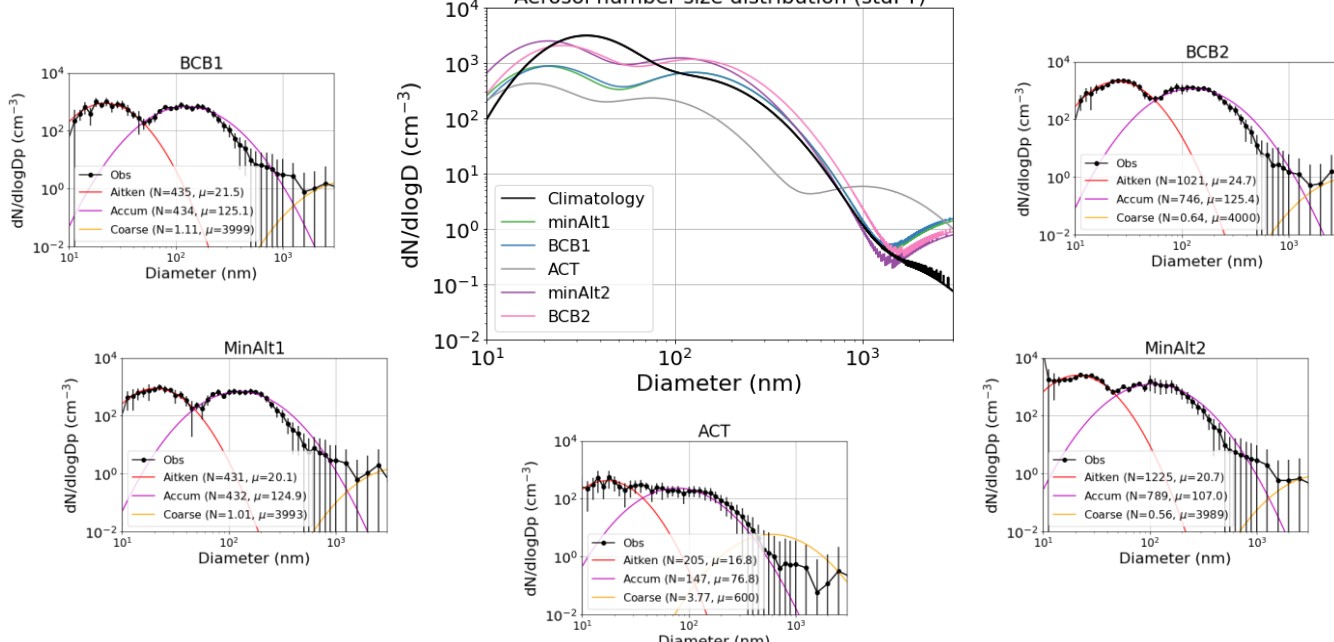

**Figure 6: (centre) Aerosol number size distribution from (black) E3SM prescribed aerosol file from climatological run near the height of simulated cloud base (~900 m AGL) and (colours) aircraft measurements averaged for each out-of-cloud flight leg fitted to 3-mode lognormal distributions. (surroundings) Mean observed aerosol number size distribution and one standard deviation (vertical lines) from each out-of-cloud flight leg and the lognormal fittings for Aitken, accumulation, and coarse modes. The fitting parameters (N in cm$^{-3}$ and μ in micrometres) are shown in the figure legends with the geometric standard deviation ($\sigma_g$) set as 1.6 for Aitken mode and 1.8 for accumulation and coarse modes. All data are converted for standard pressure (1013.25 hPa) and temperature (273.15 K) conditions.**

The fitted lognormal parameters from aircraft measurements are used to calculate and replace the variables in the E3SM

prescribed aerosol input data. The averaged chemical component fractions below 1.5 km from E3SM aerosol climatology are

used to partition the measured aerosol number size distribution so they all have the same fraction of aerosol components. The

sensitivity to different aerosol chemical compositions will be discussed in Sect. 4.2, while in this section we only focus on

how aerosol number concentration impacts clouds in E3SM-SCM. The prescribed aerosol concentration has no information

of variation with height. This height-independent assumption is usually used in SCM configurations with observed aerosols

(e.g., Liu et al., 2007; Klein et al., 2009; Liu et al., 2011), assuming that only cloud-base aerosols are involved in the cloud

droplet nucleation processes (e.g., Liu et al., 2011). Nonetheless, we also conduct a sensitivity study on aerosol vertical

distributions in Sect. 4.3.



Figure 7 (a-f) shows the vertical profiles of aerosol and cloud properties from the E3SM-SCM aerosol sensitivity simulations
between 15:00 and 16:00 UTC. The large variation of CCN number concentration has a very small impact on the cloud
fraction and in-cloud LWC. Instead, it mainly impacts the cloud droplet number and size: more CCN number concentration
leads to more $N_d$ and smaller droplet size. However, all the simulations underestimate $N_d$ compared to the aircraft
measurements. As seen in Fig. 8, the gamma distribution assumption of the cloud droplet spectrum in MG2 generally
captures the observed droplet size distribution and reproduces well the mean droplet size, but fails to reproduce the observed
peak of $N_d$ at all three heights. A similar sharp peak of $N_d$ around 10 to 20 µm was also observed by aircraft over the
Southern Ocean and the model with the same MG2 microphysics scheme underestimated $N_d$ in a similar way (Gettelman et
al., 2020). Since observed aerosols are used to drive the SCM simulations, the underestimation of $N_d$ indicates that the
turbulence in SCM is likely too weak that produces lower supersaturation thus cannot activate enough aerosols into cloud
droplets. This is confirmed by the evidence that E3SM-SCM underestimates vertical velocity variance when compared to the
Falcon measurements (Fig. 5e), and is a general bias seen in the entire ACTIVATE campaign (Brunke et al., 2022).





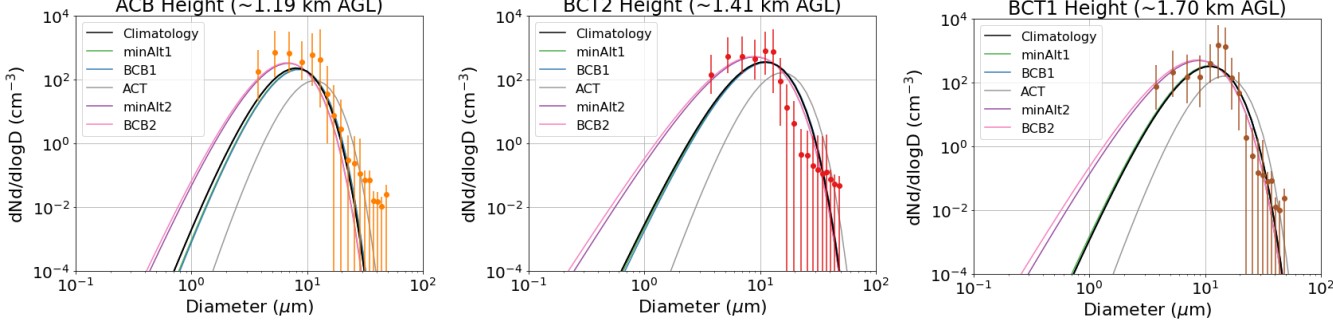

**Figure 7: Vertical distributions of (a) CCN number concentrations at 0.1% and (b) 0.5% supersaturation, (c) cloud fraction, (d) in-cloud LWC, (e) $N_d$, (f) $R_{eff}$, and (g) cloud water tendency from the conversion-to-precipitation processes in E3SM-SCM simulations with different aerosol specifications. Aircraft measurements of cloud microphysical properties overlaid are the same as in Figure 5.**





**Figure 8: E3SM-SCM simulated cloud droplet size distribution at the height of three in-cloud flight legs: (ACB: ~1.20 km, BCT2: ~1.44 km, BCT1: ~1.74 km). The dots and error bars represent aircraft measurements at the corresponding flight legs and 5th and 95th percentiles.**

The strong impact of aerosol number size distribution on cloud microphysical properties (number, size) in SCM indicates that E3SM shows a strong Twomey effect (Twomey, 1977, 1959). The change of $N_d$ is tightly related to the change of CCN number concentration (Fig. 9). A recent study of long-term E3SM simulation over the eastern North Atlantic suggests that the $N_d$ susceptibility (i.e., $\frac{d\ln N_d}{d\ln CCN}$ relationship) in E3SM may be too strong comparing to observations (Tang et al., 2023). Previous studies showed that $N_d$ is also impacted by other factors such as updraft velocity (e.g., Kirschler et al., 2022; Chen et al., 2016), which indicates a potential need of examining updraft velocity in E3SM in the future. The surface downward shortwave flux is largely impacted by the change of cloud droplet number and size due to different aerosol specifications (Fig. 10c), with the differences reaching up to 100 W m$^{-2}$ during the analysis period (15:00 – 16:00 UTC).

In contrast to the strong Twomey effect, the weak impact of aerosols on cloud macrophysical properties (cloud fraction, total water content) indicates a very weak LWP adjustment in E3SM. The LWP susceptibility $\frac{d\ln LWP}{d\ln N_d}$ is almost zero (Fig. 9c). The slightly positive slope is likely due to the suppression of precipitation processes (Fig. 7g) when cloud droplet sizes decrease in responses to more aerosol particle and cloud droplet numbers. However, the magnitude of precipitation rate change is so small that it can barely change the overall LWP and surface precipitation (Fig. 10). This weakly linear $\frac{d\ln LWP}{d\ln N_d}$ relation in the E3SM-SCM simulations is different with the non-linear $\frac{d\ln LWP}{d\ln N_d}$ relation seen in the long-term E3SM GCM run (Tang et al., 2023). Whether this weak $\frac{d\ln LWP}{d\ln N_d}$ susceptibility is a case-specific feature, the SCM simulation constrained by large-scale forcing has a lack of a feedback mechanism, or there is a large LWP – $N_d$ covariance with different thermodynamic conditions warrants future studies with more SCM cases or long-term simulations.

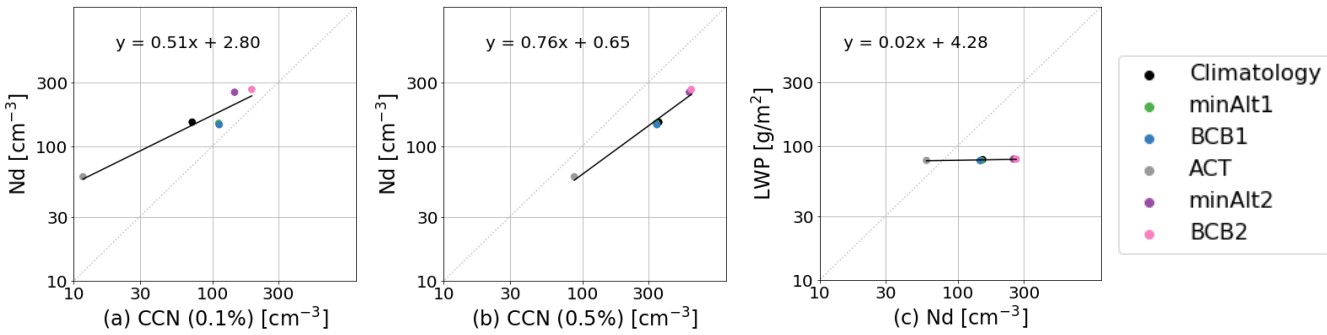

**Figure 9: Scatter plot between simulated $N_d$ and CCN at two different supersaturations and between LWP and $N_d$. The linear fit equations representing $\frac{d\ln N_d}{d\ln CCN}$ and $\frac{d\ln LWP}{d\ln N_d}$ are noted in each panel.**





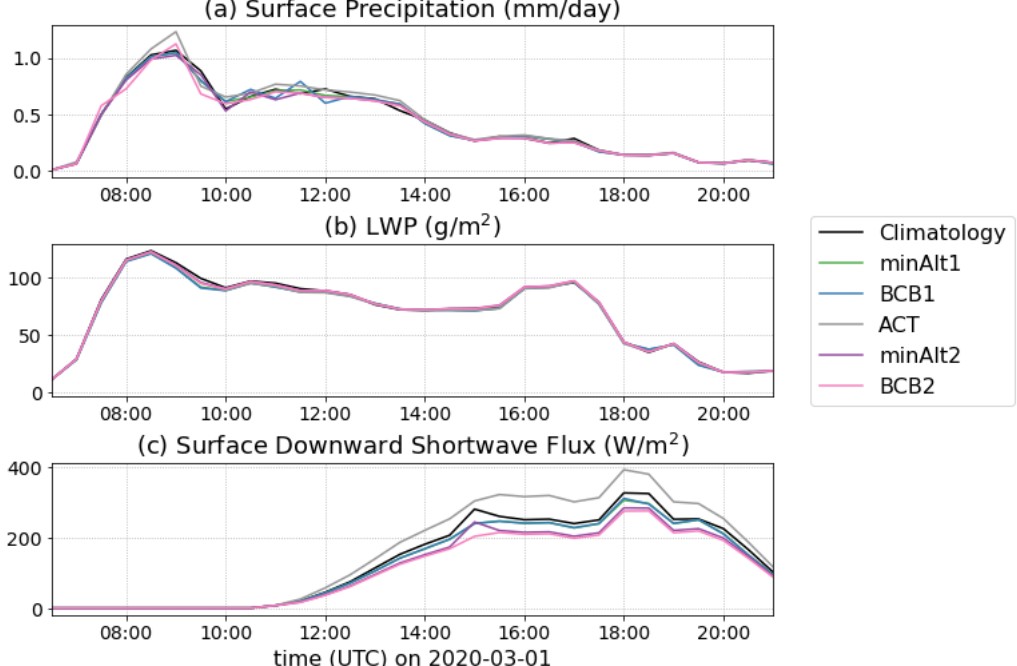

**Figure 10: Time series of (a) surface precipitation, (b) LWP, and (c) surface downward shortwave flux from E3SM-SCM simulations with different aerosol specifications.**

**4.2 Sensitivity to different aerosol composition**

Aerosol chemical composition is an important property that determines the aerosol hygroscopicity (κ) and further impacts the likelihood of aerosols serving as CCN and being activated into cloud droplets. In E3SM, the overall κ is calculated assuming internal mixing of aerosol species within each mode and external mixing among modes (Liu et al., 2012; Liu et al., 2016). Although aerosol chemical composition also impacts the overall size distribution (Shrivastava et al., 2017), this mechanism is not implemented in the current E3SM. In this section, we investigate the differences of aerosol composition used in E3SM and observed by Falcon aircraft measurements, and further test the sensitivity of simulated clouds to aerosol composition using simulated and observed values and assuming a few extreme conditions, focusing on the change of hygroscopicity.

Figure 11a shows the aerosol mass concentrations for each component in the E3SM aerosol climatology. Most of the aerosols are concentrated within the boundary layer below 1 km, with the Aitken and accumulation modes dominated by sulphate, and the coarse mode dominated by sea salt aerosols. Figures 11 (b-f) all use the same observed aerosol number size distribution, fitted from the BCB2 flight leg, but combined with different aerosol component fractions. The setting of "E3SM fraction" uses aerosol composition from E3SM prescribed aerosols at the level closest to the BCB2 leg (near ~900 m AGL). The "BCB2 fraction" uses aerosol composition from the AMS measurements at the BCB2 leg. Among the five components




in AMS measurements (Table 2), sulphate (SO4) and organics are the two dominated species observed during ACTIVATE
(Dadashazar et al., 2022a). They are also the only two species specified in E3SM, with assumptions of the composition of
organics. Here we assume all AMS measured organics are secondary organic aerosols (SOA), then calculate new aerosol
concentrations using the observed mass fraction of SO4 and SOA while keeping the fraction of other species the same in
E3SM. It can be seen that the aircraft measured SO4:SOA ratio is about 1:1 in mass, much smaller than in the E3SM
climatology. This change results in a reduction of κ value from 0.46 to 0.31 (Table 2) as the hygroscopicity of SOA is much
smaller than SO4.

Three other idealized aerosol settings in extreme conditions are provided for the purpose of sensitivity test. The first one,
"Lowest κ", is the option to use the lowest hygroscopicity species in each mode. The second option assumes all aerosols are
SO4 aerosols and the third one assumes all sea salt aerosols. The corresponding aerosol fraction in each mode and the overall
κ values are given in Table 2. The "Lowest κ" option has an extremely low κ value of $10^{-10}$ in the accumulation mode, while
the "all seasalt" option has a large κ of 1.16. The other options have κ values varying from 0.3 to 0.5.

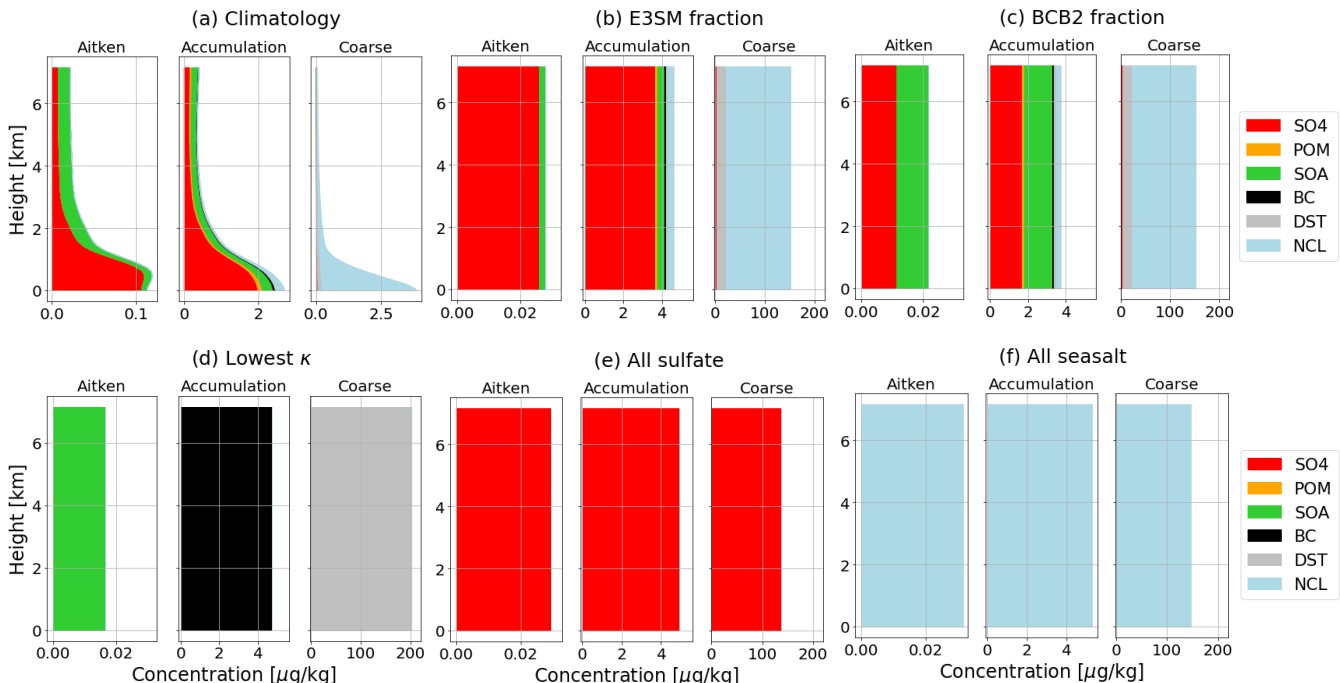


**Figure 11: Different settings of aerosol mass concentration for each component used in E3SM from (a) climatology from E3SM**
**GCM output, (b) applying composition fraction from E3SM climatology aerosols at the height of BCB2 flight leg, (c) using**
**observed fraction of sulphate and organics (assuming SOA) from the BCB2 flight leg, (d-f) assuming all aerosols are the lowest**
**hygroscopicity species ("Lowest κ") in that mode, sulphate and sea salt aerosols, respectively. Note the different x-axis in panel (a)**
**and (b)-(f). In (b)-(f), the aerosol number size distributions are from aircraft measurements in the BCB2 flight leg and assuming**
**no vertical variation. Notation of aerosol species: SO4: sulphate, POM: primary organic matter, SOA: secondary organic aerosols,**
**BC: black carbon, DST: dust, NCL: sea salt.**



**Table 2: Fraction of aerosol species in each mode (Aitken/accumulation/coarse modes) specified in five sensitivity tests. "-" means the species is not accounted for in the mode.**

| Sensitivity test | SO4 | POM | SOA | BC | DST | NCL | $\kappa^*$ |
|---|---|---|---|---|---|---|---|
| **E3SM fraction** | 0.89/0.75/0.02 | -/0.04/- | 0.11/0.12/- | -/0.02/- | -/0.02/0.09 | 0.00/0.05/0.88 | **0.46** |
| **BCB2 fraction** | 0.39/0.34/0.02 | -/0.04/- | 0.61/0.53/- | -/0.02/- | -/0.01/0.09 | 0.00/0.05/0.88 | **0.31** |
| **Lowest κ** | 0/0/0 | -/0/- | 1/0/- | -/1/- | -/0/1 | 0/0/0 | **$10^{-10}$** |
| **All sulphate** | 1/1/1 | -/0/- | 0/0/- | -/0/- | -/0/0 | 0/0/0 | **0.507** |
| **All seasalt** | 0/0/0 | -/0/- | 0/0/- | -/0/- | -/0/0 | 1/1/1 | **1.16** |

*: κ is calculated from the accumulation mode.

The different aerosol hygroscopicity results in different CCN number concentrations (Fig. 12a and 12b). As SS increases, the critical diameter determining CCN number concentration decreases and becomes less sensitive to hygroscopicity. Therefore, except the "Lowest κ" sensitivity run in which the CCN number concentration is almost zero, the relative difference of CCN number concentration with different aerosol composition settings is smaller for 0.5% SS than 0.1% SS. $N_d$ and $R_{eff}$ are less sensitive to aerosol hygroscopicity ranging from 0.31 to 1.16 compared to CCN number concentration, and cloud fraction and LWC vary even less. The only outlier is the "Lowest κ" option with extremely low hygroscopicity. In this case the extremely low CCN and $N_d$ number concentration (but not zero, as the E3SM model sets a lower limit of $N_d$ = 10 cm$^{-3}$ when cloud exists) lead to about doubled droplet size (Fig. 12f). Therefore, it has a much stronger surface downward shortwave radiation (Fig. 13c). The much larger droplet size also contributes to more precipitation conversion (Figs. 12g and 13a) and depletion of cloud liquid water (Fig. 13b). However, the impact is still very weak and the estimated LWP susceptibility $\frac{d\ln LWP}{d\ln N_d}$ is 0.02 (Fig. 14c).







**Figure 12: Same as Figure 7 but for E3SM-SCM simulations with different aerosol compositions and same aerosol number concentration (except Climatology) from BCB2 measurements.**




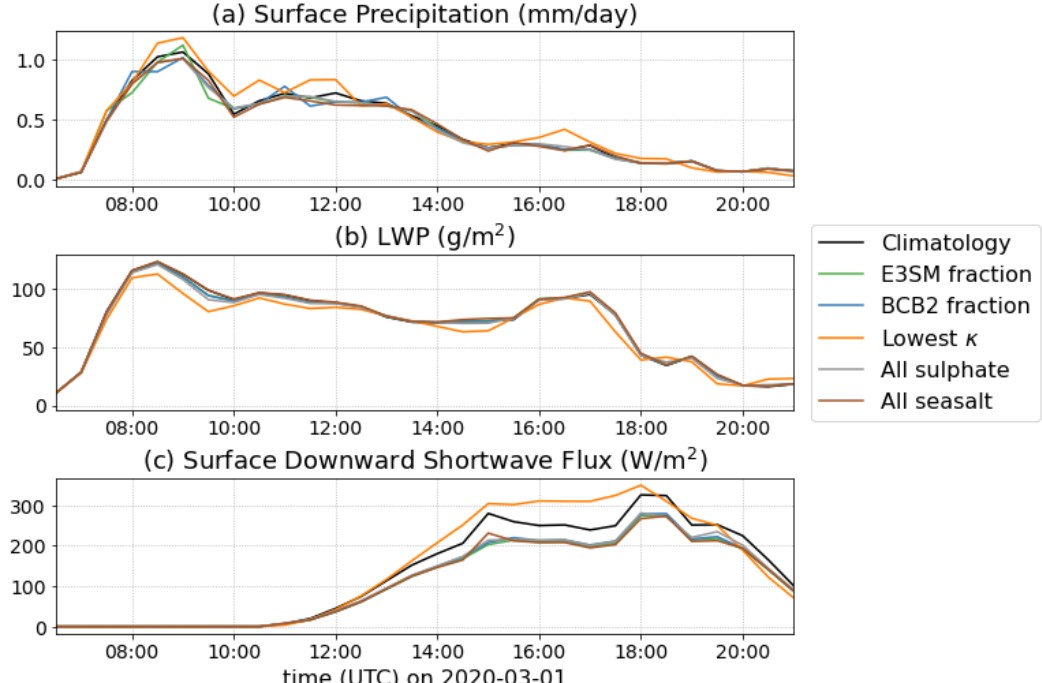

**Figure 13: same as Figure 10 but for E3SM-SCM simulations with different aerosol compositions.**

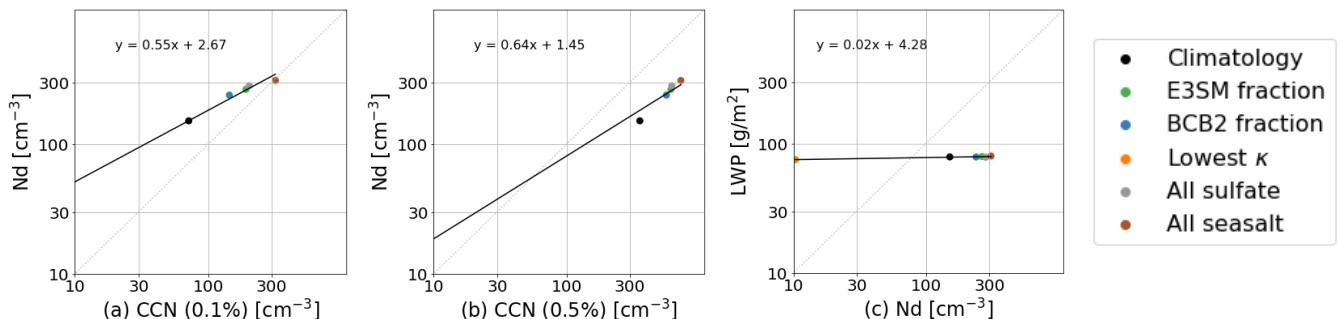

**Figure 14: same as Figure 9 but for E3SM-SCM simulations with different aerosol compositions.**

**4.3 Sensitivity to aerosol vertical distribution**

In many previous modelling studies using observed aerosols, usually only one set of aerosol parameters (i.e., particle number size distribution and composition) was given to the model regardless of the vertical distribution (Liu et al., 2011; Liu et al., 2007; Klein et al., 2009; Lebassi-Habtezion and Caldwell, 2015; Li et al., 2023). The observed aerosol information is usually taken from in-situ measurements below cloud base (e.g., Liu et al., 2011; Li et al., 2023), assuming that hygroscopic aerosol particles are readily activated into cloud droplets in the saturated air driven by updrafts. However, as aerosol concentration usually decreases with height, the aerosol vertical distribution may be changed by in-cloud scavenging, horizontal transport and vertical mixing, which further affect the cloud microphysical properties (e.g., Lin et al., 2023; Zhang et al., 2021;





Kirschler et al., 2022). Indeed, the secondary activation of aerosols above cloud base has been shown to have a significant impact on aerosol convective removal and vertical transport (Wang et al., 2013; Wang et al., 2020). Here we perform a sensitivity study to investigate the impact of aerosols at different vertical levels on E3SM-SCM simulated clouds, and further assess the impact of aerosol vertical distribution on clouds, comparing to results from the simulations with constant vertical aerosol concentration.

In this set of sensitivity tests, we prescribe aerosols from BCB2 flight leg only for a single model layer, with all other layers being aerosol-free. We also perform a simulation with idealized aerosol vertical distribution, where aerosol number concentration decreases linearly from 1 km to 2 km AGL (approximately within the cloud layer) to 10% of its boundary-layer value. Figure 15 shows the vertical profiles of the simulation results. With a prescribed aerosol configuration, the cloud activation process only takes the aerosol information in that layer. However, when aerosol particles are activated into cloud droplets, they are redistributed vertically via vertical transport and sedimentation. The aerosols below cloud base and above cloud top do not participate in the cloud activation process, with $N_d = 10$ cm$^{-3}$ (the low cut-off value) and large R$_{eff}$ similar to the "Lowest κ" results in Fig. 12. Aerosols within the "Cloud Base" and "In-Cloud" layers contribute to about 30% to 40% of $N_d$ activated in the "Constant" aerosol run throughout the simulated cloud layer. The "Cloud Top" aerosols mainly contribute to $N_d$ at the cloud-top layer, with a few droplets falling to lower levels causing a reduction in droplet size (Fig. 15f). The "Idealized" aerosol profile generally captures the vertical distribution of aircraft measured CCN (Fig. 15b), albeit aircraft measured CCN is overall smaller near the cloud base, likely due to the aerosol scavenging process. Although the decrease of aerosols is 90% at the cloud top, the reduction of $N_d$ in the "Idealized" case is only 20% to 30% less than the "Constant" case (Fig. 15e). Since E3SM-SCM underestimates $N_d$ in this case, it is difficult to demonstrate the value of adding aerosol vertical variation. Moreover, the prescribed-aerosol setting in E3SM-SCM limits its ability to study ACI. An interactive aerosol configuration with vertical transport and other processes such as dry and wet deposition enabled is needed to further understand the impact of aerosol vertical distribution on clouds and ACI.



429



**Figure 15: Same as Figure 7 but for E3SM-SCM simulations with (gray): constant aerosol number concentration (per kg air), (black): idealized aerosol profile with number concentration decreasing from 1 to 2 km AGL to 10% of the MBL concentration, and (colours): aerosols in a single layer only. Aircraft measured CCN number concentrations for SS between 0.45% and 0.55% are overlaid in (b). Aerosol number concentration is from aircraft measurements in the BCB2 leg while aerosol composition is from E3SM climatology at the BCB2 leg height.**

## 5 Summary and Discussion

Current ESMs remain largely uncertain in simulating MBL clouds, and ACI related to MBL clouds have been underexplored over the WNAO. With the recent ACTIVATE field campaign conducted over WNAO collecting in-situ and remote-sensing measurements using dual aircraft flying simultaneously, we perform a model intercomparison and sensitivity study for a selected CAO case to understand the complex aerosol-cloud interactions related to MBL clouds over WNAO.




A unique feature of this study is the multi-scale model intercomparison using SCM, CRM and LES models, which provides
a comprehensive process-level understanding of ACI in more details compared to individual models. We conducted E3SMv2
simulations in the SCM mode, and compared with two WRF model configurations at LES and CRM resolutions,
respectively. Overall, the three models all capture the MBL cloud properties, while the E3SM-SCM underestimates cloud
droplet number concentration and overestimates droplet size. This is partly due to the relatively low concentration of
prescribed aerosols from the E3SM climatology compared to the observation in this case, and partly due to underestimated
updrafts that cannot activate enough aerosol particles into cloud droplets. Note that some parameters in E3SMv2 were tuned
to improve the overall performance of subtropical stratocumulus clouds (Ma et al., 2022), but turbulence over the WNAO
region is weakened comparing to the pre-tuning version (close to E3SMv1) even in a long-term GCM run (Brunke et al.,
2022). The evaluation of SCM simulations against the ACTIVATE measurements is helpful for understanding and
improving turbulence representation over this region.

Among the three models, E3SM-SCM and WRF-LES are driven by the same large-scale and surface forcings derived from
ERA5 reanalysis, while the WRF-CRM model is run as a regional model with nested-domains. However, only the WRF-
CRM reproduces the characteristics of cloud rolls in this cold-air outbreak case (Chen et al., 2022). With the same large-
scale and surface forcings from WRF-CRM, which has weaker subsidence and stronger low-level cold and dry air advections
than ERA5 forcings, the E3SM-SCM and WRF-LES produce much thicker clouds than WRF-CRM. This indicates that a
proper match of large-scale dynamics, sub-grid scale parameterization, and model configurations is needed to obtain optimal
model performance.

Several sets of sensitivity experiments are conducted to examine ACI by changing the prescribed aerosol number size
distribution and aerosol composition in E3SM-SCM. Aircraft measurements at different heights are used to provide
constraints of the aerosol perturbation. Changing aerosol number size distributions dramatically alters the CCN number
concentration, thus largely impacts cloud droplet number concentration and size, further influencing the cloud radiative
effect. However, changing aerosol composition only shows dramatic impacts in the extremely low hygroscopicity ($\kappa$) setting,
where there are only very few aerosols being activated into very large cloud droplets. Changing the overall $\kappa$ from 0.31 to
1.16 has a smaller impact on cloud microphysical properties. Worth noting, the impact of aerosol composition to CCN
concentration and cloud microphysics can be larger than shown here as it may also change the aerosol size distribution
(Shrivastava et al., 2017).

In contrast to the clear Twomey effect, the cloud fraction and water content are barely impacted by aerosol perturbations,
with a very weak $\frac{d\ln LWP}{d\ln N_d}$ susceptibility of 0.02. The slight positive LWP adjustment is most likely due to the rain suppression
effect (Albrecht, 1989) for smaller cloud droplets. This contradicts the non-linear V-shape $\frac{d\ln LWP}{d\ln N_d}$ curve shown in the long-





term E3SM GCM run over the Eastern North Atlantic Ocean (Tang et al., 2023; Varble et al., 2023). Whether this weak positive LWP susceptibility is a case or location specific feature and whether SCM can reveal the same cloud susceptibility as GCM does are subject to further study.

We also performed a sensitivity study to test the impact of aerosol vertical distribution on cloud simulations. Due to the prescribed-aerosol configuration in E3SM-SCM, only aerosols at cloud levels can be activated. Adding aerosol vertical variation (i.e., decreasing concentration with height) reduces the simulated $N_d$ as there are lower concentrations of aerosols in cloudy layers than below cloud base. However, this may not be necessarily better than vertically constant aerosols obtained below cloud base, because there is no treatment of vertical transport of aerosols in the SCM configuration. A more comprehensive SCM simulation with complete vertical transport and other aerosol processes is needed to better simulate ACI and connect field measurements and process-level models with global models.

**Data Availability**

The ACTIVATE aircraft data and GOES-16 satellite data are available from the NASA ACTIVATE project website (https://asdc.larc.nasa.gov/project/ACTIVATE, DOI: 10.5067/SUBORBITAL/ACTIVATE/DATA001). ERA5 reanalysis data are available from the Copernicus Climate Change Service Climate Data Store (CDS) (Hersbach et al., 2023a, b).

**Code Availability**

The E3SMv2 model is available from the U.S. Department of Energy at https://doi.org/10.11578/E3SM/dc.20210927.1 and the SCM scripts are revised from the E3SM SCM library (https://github.com/E3SM-Project/scmlib). The WRF community model is available from the National Center for Atmospheric Research (NCAR) at http://www2.mmm.ucar.edu/wrf/users/.

**Author contribution**

ST and HW designed the conceptional ideas. AS, HW, and XZ performed the mission planning and supervision. EC, KT, LZ and CV participated in mission operation and data curation. ST conducted the SCM simulations, XYL conducted the WRF-LES simulations, and JC conducted the WRF-CRM simulations. ST performed the analysis and prepared the original manuscript. All co-authors contributed to the reviewing and editing of the manuscript.

**Competing interests**

AS and HW are members of the editorial board of Atmospheric Chemistry and Physics. Other authors declare that they have no conflict of interest.





## Acknowledgments

**Acknowledgments**
This work was supported through the ACTIVATE Earth Venture Suborbital-3 (EVS-3) investigation, which is funded by
NASA's Earth Science Division and managed through the Earth System Science Pathfinder Program Office. The Pacific
Northwest National Laboratory (PNNL) is operated for the U.S. Department of Energy by Battelle Memorial Institute under
Contract DE-AC0576RLO1830. The simulations were performed using resources available through Research Computing at
PNNL. University of Arizona investigators were funded by NASA grant no. 80NSSC19K0442. CV was funded by the
German Research Foundation DFG within projects SPP-1294 HALO under Vo1504/7-1 and Vo1504/9-1.

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
