# Peer review of "Understanding Aerosol-Cloud Interactions in a Single-Column Model: Intercomparison with Process-Level Models and Evaluation against ACTIVATE Field Measurements"

_EGUsphere, 2023_

## Author Comment (AC1)

**Reviewer #1:**

**The authors perform several modeling experiments based on a flight during the ACTIVATE campaign. First, they compare the results of range of models (WRF as CRM, WRF as LES, and E3SM as SCM) in reproducing properties of the clouds and boundary layer observed during the campaign. Then, the authors demonstrate that it is necessary to use identical simulation forcings between models to obtain comparable results between the models. The authors then perform a set of experiments to test the sensitivity E3SM-SCM to the treatment of aerosols in the model (size distribution, species, and vertical distribution).**

**Overall, there were several points in the manuscript that left me confused about what exactly was being done with the simulations (General Comment 1). I also find it difficult to understand the usefulness of these results because only one hand-picked case is used in these experiments (General Comment 2). Finally, I question whether the E3SM-SCM is appropriate for these science questions, as it appears to not represent the effects of aerosol scavenging, an important mechanism in aerosol-cloud-interactions (General Comment 3). To address these issues, I expect that major revisions are necessary before publication.**

Thank you for reviewing the manuscript and providing valuable comments. We have revised the manuscript thoroughly to answer your questions. Some major changes are listed below:

1. We revised the title to "Understanding Aerosol-Cloud Interactions Using a Single-Column Model for a Cold-Air Outbreak Case during the ACTIVATE Campaign" to emphasize the focus of this study (i.e., a cold-air outbreak case) and deemphasize the intercomparison with CRM and LES models.
2. We moved the results of using CRM forcing and the sensitivity tests of aerosols at different vertical layers into the supplement.
3. We added the sensitivity of SCM results to vertical velocity variance, which shows that the underestimation of cloud droplet number concentration in E3SM is partly due to an underestimation of aerosols and partly due to an underestimation of turbulence strength.
4. We added a section to further investigate the LWP susceptibility, including more discussion on SCM behavior and differences from GCM results.

Although we have decided not to add more cases to the manuscript (see our response to General Comments 1 and 2), we did revise the analysis and associated discussion thoroughly to demonstrate our findings from this typical cold-air outbreak case, and provided discussions on designing long-term simulations for statistical analysis (as opposed to the case study approach).

In the SCM configuration with prescribed aerosols, aerosol properties are fixed so we can only use the SCM configuration to study the impact of aerosols on clouds, but not the microphysical or dynamical feedback to aerosols. However, the prescribed or observed aerosols may have considered such feedback in the full GCM simulation or real-world atmosphere. We have added this point in the model introduction:

*"we use a "prescribed-observed" hybrid method in this study, in which we replace the prescribed aerosol input data with aircraft-measured aerosols or idealized conditions. Note that we may only study the impact of aerosols on clouds in this configuration, but not the interactive microphysical and*

*dynamical feedback to aerosols, as model representations of aerosol sink and source processes such as emissions, scavenging, and deposition are disabled in this configuration."*

Below please find our point-by-point responses to the specific comments.

**General Comments**

**Overall, I was confused by the experimental design at several parts of the manuscript. I think the manuscript would benefit from more details for each experiment about exactly how many simulations are performed and with what model/forcing differences. Some of the Specific Comments below address this issue.**

**There is only one case here and the choice of this single case is not motivated in the text. The authors correctly identify this as a limitation to the interpretation and application of their results (Line 475 and elsewhere). Utilizing only a single hand-picked case both introduces bias and limits the application of these findings to general ACI. I am not convinced these results can be impactful to the community due to this limitation. By what criteria was this case chosen instead of the other eleven "process study" cases? Most of the experiments in this work are performed with a computationally cheap single column model. It seems that more cases could be added, thereby increasing representativeness of the results without considerably increasing complexity or computational cost.**

Thank you for the comments. We have added a few lines of text to address this concern (see our reply to Comment "Section 2.1" below).

We agree that one single case is limited to generate a robust or general conclusion on ACI. We did attempt to consider other "process study" cases during the ACTIVATE field campaign. However, in practice, there are a few issues preventing us adding more cases in this ACI study of CAO over the western North Atlantic:

1. Aerosol-cloud interactions over the western North Atlantic are highly dependent on synoptic conditions. This study is mainly focused on cold-air outbreak conditions during the ACTIVATE campaign. For the limited number of process-study cases sampled during the ACTIVATE campaign, most of them are for summer cumulus clouds (e.g., Sorooshian et al., 2023; Li et al., 2024). Therefore, we don't have many cases to choose from if we want to evaluate the SCM results against extensive ACTIVATE observations and compare with fine-resolution process-model results.
2. The E3SM SCM fails to reproduce the cloud evolution of the other ACTIVATE CAO case. We conducted E3SM-SCM simulations for the CAO case on February 28, 2020. However, the SCM fails to reproduce the observed persistent MBL clouds and the cloud growth later in the day (see the plot below). As ACI metrics and sensitivity to aerosols are dependent on cloud and thermodynamic conditions, the failure of reproducing cloud evolution makes it questionable to study ACI in SCM for this CAO case.

[Figure]

Figure R1: cloud fraction evolution from E3SM-SCM, WRF-LES, and ERA5 for the 28 February case.

Part of the reason for choosing the 1 March 2020 case is the well-represented CAO synoptic conditions (and forcings from reanalysis) and the good performance of MBL clouds in other models. As seen in the previous studies with WRF-CRM and WRF-LES models (Chen et al., 2022; Li et al., 2022, 2023), a single well-simulated case may still provide useful information on understanding the dynamics, thermodynamics, and aerosol-cloud-meteorology interactions over this region. We have also revised the structure of this paper and added a few more sensitivity tests and analyses to further understand the behavior of model physics in E3SM-SCM. On the other hand, we do have a plan of running long-term SCM simulations for statistical analysis over the ACTIVATE domain. However, including long-term statistical analysis in this CAO case study would defeat the purpose of evaluating the SCM against field observations and intercomparing SCM with CRM/LES results, so we decided to keep this paper as a single CAO case study and focus on the long-term analysis in a follow-up study.

**The authors find that different aerosol concentrations strongly affect Nd but have minimal effects on the macroscale simulation properties (LWP, cloud fraction, surface rain). Only changing the hygroscopicity to 10^-10 has any effect on the macroscale properties. I wonder about the cause of this lack of macroscale aerosol effects. I wonder if the limited scope (a portion of a single hand-chosen flight of a much larger and long-lasting campaign), the model characteristics (formulations, simulation duration), or some other factor might be avoiding macroscale ACI. Please address this issue in the text.**

For this cold-air outbreak case, the strong subsidence, cold-air advection, and surface turbulent heat fluxes take control of the cloud formation and macrophysical properties (e.g., LWP and cloud fraction). Aerosol effects mainly alter cloud microphysical properties, such as cloud droplet number and size, which are shown to have a very minor impact on cloud LWP. By no means can the findings be generalized to different cases in the ACTIVATE domain. We believe that under the synoptic conditions with weaker large-scale forcings and/or stronger precipitation, the impact of aerosols on cloud macrophysical properties may be stronger. We have added the following text for clarity:

*"In the CAO case, LWP and other cloud macrophysical properties are likely determined by the strong dynamical and thermodynamical controls (e.g., strong cold-air advection, surface turbulent heat fluxes, and subsidence in Fig. 2). The change of aerosols mainly impacts cloud microphysical properties through altering cloud droplet number and size, which is shown to have a minimal effect on cloud LWP. We believe that under the synoptic conditions with weaker large-scale forcings and/or stronger precipitation, aerosol effects on cloud macrophysical properties may be stronger."*

**Minor Comments**

**Line 19: "as good" should be replaced with "as well".**

We have revised this sentence as:

*"Results show that E3SM-SCM well reproduces the macrophysical property of post-frontal boundary layer clouds for a cold-air outbreak (CAO) case."*

**Section 2.1: Please explain why this case was chosen (other than it was used in previous publications by the authors). Why not include any of the other 12 "process study" flights?**

As we explained in the response to your general comments above, aerosol-cloud interactions over the western North Atlantic are highly dependent on synoptic conditions, and this study is mainly focused on cold-air outbreak conditions during the ACTIVATE campaign. There were other CAO cases observed during ACTIVATE, but E3SM SCM fails to reproduce the cloud evolution in another CAO case we tested. We have added the explanation in the last paragraph in the introduction, right before Section 2:

*"In this study, we focus on SCM simulations of the same CAO case as that being investigated in the CRM/LES studies (Chen et al., 2022; Li et al., 2022; Li et al., 2023). We tried a few other CAO cases observed during the ACTIVATE campaign, but the SCM cannot produce the observed boundary-layer structure and cloud evolution in those cases, likely due to weaker CAO forcings and not as well-defined large-scale boundary conditions for the SCM. It is critical to have well simulated clouds for the aerosol-cloud-interaction sensitivity tests. Therefore, our study is limited to this single case."*

**Line 120: Please provide a description of the Xie et al. (2019) modification. This addition can be very brief.**

The modification from Xie et al. (2019) includes two parts: 1) using dynamical CAPE as a triggering threshold to bring in large-scale control of convection initiation, and 2) relaxing the limit of parcel lifting level to allow above-PBL elevated convection to be triggered. As the deep convective scheme is not triggered for this MBL case and is not key to the ACI analysis in this paper, we just add a brief description:

*"…with the modification in convective trigger from Xie et al. (2019) to improve the diurnal cycle of precipitation"*

**Line 125: "use" should be "using" and "has" should be "have". There are a few other examples of small mix-ups like this.**

Thank you for the comments. We have corrected them and carefully read through the revised manuscript to eliminate writing errors.

**Fig 2: Please stretch this figure vertically so the reader can examine changes with height.**

Revised as suggested.

**Fig 4: The King Air observations are limited in time. It would be helpful to include GOES-16 ABI retrievals for Cloud Top Height, if they are available, as the authors have already done for Total Liquid Water Path.**

Added as suggested. However, the satellite retrieved cloud top height (including both GOES-16 and CERES data) is about 1 km higher than in the aircraft measurements and the model simulations. We have also added the following text to discuss this issue:

*"Figure 4a shows the time series of cloud top height compared with GOES-16 satellite measurements and HSRL-2 measurements from the King Air aircraft. It should be noted that although both are measured from above the cloud, the satellite-measured cloud top height is about 1 km higher than the aircraft lidar measurement. As this is only a case study, we do not attempt to address whether the satellite measurement has any systematic bias. HSRL-2 detects the top of each individual cloud, which is usually lower than or, at best, equal to the highest cloud top within the area. Therefore, we only compare model results with the highest values of the HSRL-2 measurements."*

**Fig 6: The ACT Coarse mode fit does not represent the observations, resulting in the appearance of a too-small and too-populated coarse mode (more than 3 times as populated as the near-surface legs!). I suggest re-doing the fits but with a minimum on the fitted mu of more than 1-2 micron. It also seems that simply removing this mode from the ACT leg could be appropriate, given the observed counts above 1 micron are considerably fewer than the below-cloud legs (log-scale currently hides this). The authors mention that the coarse mode probably doesn't exert a large effect on the simulations (and I agree) but these fits should be recalculated to avoid misleading the reader in Figure 6. Doing so requires repeating the sensitivity experiments, but the SCM framework permits computationally cheap simulations.**

We have changed the upper and lower bounds of the fitting parameters and re-did the fitting for ACT aerosols. All other simulations and plots are revised accordingly.

**Figures 7-8 and related text: I am confused by these sensitivity experiments and need more description. Are they only 1 hour simulations or are they for the full 14 hours but only analyzed during the 15-16 UTC? I don't even understand how many E3SM-SCM simulations contributed to Fig 8. Are the aerosol concentrations measured during the different flights used throughout the column but adjusted to retain a desired dependence of total number with height? Please elaborate the experiment design and exactly what is being shown in Figures 7 and 8.**

We have revised the text and figure caption as follows to avoid such confusion:

Text:

*"All simulations are run from 06:00 to 21:00 UTC, the same as the previous simulations described in Sect. 3. To compare with aircraft measurements, we average the simulations between 15:00 and 16:00 UTC (aircraft sampling time) and plot the vertical profiles in Fig. 7. The large variation of CCN number concentrations has a very small impact on the cloud fraction and in-cloud LWC. Instead, it mainly impacts the cloud droplet number and size: more CCN leads to more $N_d$ and smaller droplet size. However, all the simulations underestimate $N_d$ compared to the aircraft measurements. A further sensitivity test shows that underestimation of both aerosol number concentration and turbulence strength contributes to the underestimation of $N_d$. When increasing vertical velocity variance to the observed magnitude and using aerosols observed below the cloud base in SCM, the simulated $N_d$ then becomes much closer to the aircraft measurements (Fig. 8)."*

Figure caption:

*"Figure 9: E3SM-SCM simulated cloud droplet size distribution at the height of three in-cloud flight legs: (ACB: ~1.20 km, BCT2: ~1.44 km, BCT1: ~1.74 km). Note that the flight leg name and height in the title above each panel specify where the cloud data are taken for the plot, while the flight leg names within each panel legend describe where the aerosol data are taken to drive the corresponding E3SM-SCM simulations. The dots and error bars represent aircraft measurements at the corresponding flight legs and 5th and 95th percentiles. "*

**Fig 7g: Please spell out the MPDW2P acronym.**

Added in the figure caption: *(MicroPhysics tendency Due to Water to Precipitation, MPDW2P)*

**Fig 10: Surface precipitation is interesting but is of course affected by sub-cloud evaporation. I think cloud-base precipitation rate would be more informative here. I suspect that the cloud-base precipitation rate is also very small, which would partially explain why aerosols seem to have little effect on macroscale quantities. Perhaps the changes in aerosol concentrations are not enough to result in macroscale changes to the simulations?**

In E3SM SCM, precipitation is predicted for cloudy grids and can fall through several model layers below cloud base, even onto the surface within one timestep (1800 s). Therefore, cloud-base precipitation rate is not predicted or diagnosed from E3SM model output, while the surface precipitation is generally lower than, but consistent, with the precipitation water within cloud after below-cloud evaporation. On the other hand, cloud properties such as LWC and cloud fraction are insensitive to the aerosol perturbations in this case, and so the choice of surface or cloud-base precipitation is not expected to affect the conclusion here.

**Line 326: Before stating any physical reasoning behind the positive slope, you should include an uncertainty range for the slope value and account for systematic uncertainty that would not be included in the regression-based uncertainty range, which assumes independent samples, etc. I suspect this slope falls within that uncertainty range, indicating that any physical reasoning is not meaningful. I don't doubt the significance of the slopes in the other Fig 9 panels, but it would be a good idea to include them, as well.**

We have now added the standard errors of slope and intercept for each linear fit in the figure caption to characterize the uncertainty range. The slope of dlnLWP/dlnNd, despite being very small, is still much greater than the uncertainty range.

**Line 331-332: This is a good point and is a major limitation to this study. Utilizing only a hand-picked case both introduces bias and limits the application of these findings to general ACI.**

As discussed above, we made a few tests but are not confident to include more CAO cases for the ACI-focus of this study. We admit the limitations of using single-case analysis for a comprehensive evaluation with observations and other models, but this is now made clear in the paper. Results and lessons learned from this case study will be used to guide long-term SCM simulations and statistical analysis. We have included the following discussions in the text:

The second paragraph in Sect. 5:

*"Further studies with more cases and associated statistical analyses are needed to verify this hypothesis."*

The first paragraph in Sect. 6:

*"… This case study with a comprehensive set of aerosol sensitivity simulations provides insight into further designing and investigation of long-term SCM simulations for statistical analysis, which is currently under consideration for a future study."*

The 4th paragraph in Sect. 6:

*"The slight positive LWP adjustment is most likely due to the rain suppression effect (Albrecht, 1989). This contradicts the non-linear V-shape $\frac{dlnLWP}{dlnN_d}$ curve shown in the long-term E3SM GCM run over the Eastern North Atlantic Ocean (Tang et al., 2023; Varble et al., 2023). Whether this weak positive LWP susceptibility is a case-specific or cloud-regime-specific feature and whether SCM can reveal the same cloud susceptibility as the full GCM does are subject to further study."*

**Fig 11 and elsewhere: Please use NaCl or Salt instead of NCL.**

Revised as suggested.

**Fig 15: I don't know what I've learned from this sensitivity experiment. Are aerosols activated only in layers with prescribed aerosols? In cloudy layers that do not have prescribed aerosols, is the Nd value set to 10/cm^3, after which time cloud droplets are transported within the cloud? If scavenging is not represented in E3SM-SCM, how are we do understand the complex ACIs?**

We have now removed this part from the main text (with only a brief mention in the Summary and Discussion section). To answer your questions, cloud droplet nucleation is calculated according to supersaturation and prescribed aerosols; however, in cloudy layers without the prescribed aerosols, $N_d$ values were set to 10 cm$^{-3}$. Once formed, cloud droplets can be transported vertically by vertical advection, turbulent transport, and sedimentation. The tendencies are calculated for each E3SM-

SCM timestep, which is 30 minutes. In the prescribed-aerosol configuration in E3SM-SCM, model representations of aerosol sources and sinks such as emissions, scavenging, and deposition are disabled. Therefore, in the SCM experiments we can only study the impact of aerosols on clouds, but not the other way around. We have added the relevant discussions to the paper (in the last section):

*"In the current SCM framework using observed aerosols, usually only one set of aerosol parameters, characterizing the spatially mean properties (i.e., particle number size distribution and composition), is fed into the model regardless of the aerosol vertical distribution (Liu et al., 2011; Liu et al., 2007; Klein et al., 2009; Lebassi-Habtezion and Caldwell, 2015; Li et al., 2023). The prescribed aerosol information based on observations is usually taken from in-situ measurements below the cloud base (e.g., Liu et al., 2011; Li et al., 2023), assuming that hygroscopic aerosol particles are readily activated into cloud droplets in the saturated air driven by updrafts. However, as aerosol concentration usually decreases with height in the lower atmosphere, regional aerosol vertical distribution may be changed by in-cloud scavenging, horizontal transport, and vertical mixing, which can further affect cloud microphysical properties by secondary activation above cloud base (Wang et al., 2013; Wang et al., 2020). We conducted a sensitivity experiment with a specified aerosol vertical distribution (Fig. S5), but the configuration of prescribed aerosols in SCM only shows the response of clouds to aerosols given at the level of cloud formation. A more comprehensive consideration of complete aerosol processes (e.g., vertical transport, scavenging, deposition, etc.) is needed to include the cloud and dynamical feedback on aerosols and better understand the aerosol-cloud interactions."*

---

## Author Comment (AC2)

**Reviewer #2:**

In this paper the authors use the ACTIVATE field campaign to compare the E3SM SCM to different flavors of WRF (in both CRM and LES mode) with regards to the simulation of clouds and boundary layer turbulence observed during the campaign. The second part of the paper focuses on a set of E3SM-SCM experiments focused on the sensitivity to treatment of aerosols in the model.

While I found aspects of this paper to be interesting, it also felt like a hodge-podge of ideas/experiments that lacked a clear unifying focus as to what the authors hoped to accomplish/address. The two distinct sections of the paper feel a bit disjointed and I think the authors could do a better job tying them together a bit more. In addition, the second part of the paper focuses on just one case from ACTIVATE to draw some conclusions. I feel this needs to be addressed by testing robustness against more flights from the ACTIVATE campaign. In addition, there were several other sources of confusion in this document that need to be addressed (please see itemized list). Overall, I feel a major revision is necessary before this article is suitable for publication.

Overall, the paper is well written enough that I understand what the authors are saying; but there are frequent typos and grammar mistakes that are distracting and needs to be addressed upon resubmission.

We thank the referee for reviewing the manuscript and providing valuable comments. We have revised the manuscript accordingly in response to the general comments here as well as the major comments from Referee #1. The manuscript has also been checked through *Grammarly* and by native English-speaking coauthors thoroughly for grammar mistakes. Below please find our point-by-point responses to your specific comments.

In the conclusions of the paper the authors state (and elude to this on other sections of the manuscript): "A unique feature of this study is the multi-scale model intercomparison using SCM, CRM, and LES models, which provides a comprehensive process-level understanding of ACI in more details compared to individual models". I'm left very confused by this statement. The CRM and LES models were only used in the first half to compare the macroscopic aspects of the SCM simulation (clouds, turbulence, etc.); I do not see how they were used to help understand ACI directly other than being used as a validation tool.

We have now revised the title and the main text to emphasize the model evaluation and process-level understanding of ACI in E3SM-SCM using ACTIVATE observations as well as the CRM/LES results for validation/comparison. The new title is "Understanding Aerosol-Cloud Interactions Using a Single-Column Model for a Cold-Air Outbreak Case during the ACTIVATE Campaign", while the comparison with CRM and LES results is now part of the SCM evaluation.

I found the comparison of E3SM-SCM to WRF interesting but was confused why the authors felt it pertinent to include the SCM and LES runs with the CRM forcing. The

conclusions they draw of "proper combination of large-scale dynamics, sub-grid parameterizations, and model configurations is needed to obtain performance…" seems like a super obvious conclusion that I'm not sure why they felt needed detailed analysis.   Unless I'm missing something I suggest that the authors remove these curves from the figures (which are too busy with these curves included) and perhaps state in a sentence or two that they explored the sensitivity to large-scale forcing.   To me this analysis and section felt like a distracting tangent.

We agree that the SCM sensitivity to the large-scale forcing is expected (without an exception for this CAO case). As we follow the suggestion to de-emphasize SCM/CRM/LES intercomparison and focus more on the ACI in SCM, we have removed the results of SCM using CRM forcing from the main text.

Page 8, line 173 the authors state "…neither resolved nor parameterized at the sub-grid scale in E3SM-SCM". What exactly is "resolved" in an SCM?   Isn't SCM just one column where all processes are parameterized?   Or am I misunderstanding something about the E3SM SCM?

What we were trying to say is that this structure is smaller than the grid size represented by the SCM. To avoid such confusion, we have revised this sentence to

*"However, this roll structure fails to be simulated in WRF-LES and is not parameterized in E3SM-SCM."*

In the second half of the document, which explores E3SM-SCM to aerosol sensitivity the authors use one case and make the statement that their conclusions "warrant more cases" to test robustness. I completely agree… why not include more cases then?

Thank you for the comments. We agree that one single case is limited to generate a robust or general conclusion on ACI. We did attempt to consider other "process study" cases during the ACTIVATE field campaign. However, in practice, there are a few issues preventing us adding more cases in this ACI study of CAO over the western North Atlantic:

1.  Aerosol-cloud interactions over the western North Atlantic are highly dependent on synoptic conditions. This study is mainly focused on cold-air outbreak conditions during the ACTIVATE campaign. For the limited number of process-study cases sampled during the ACTIVATE campaign, most of them are for summer cumulus clouds (e.g., Sorooshian et al., 2023; Li et al., 2024). Therefore, we don't have many cases to choose from if we want to evaluate the SCM results against extensive ACTIVATE observations and compare with fine-resolution process-model results.
2.  The E3SM SCM fails to reproduce the cloud evolution of the other ACTIVATE CAO case. We conducted E3SM-SCM simulations for the CAO case on February 28, 2020. However, the SCM fails to reproduce the observed persistent MBL clouds and the cloud growth later in the day (see the plot below). As ACI metrics and sensitivity to aerosols are dependent on cloud and thermodynamic conditions, the failure of reproducing cloud evolution makes it questionable to study ACI in SCM for this CAO case.

[Figure]

Figure R1: cloud fraction evolution from E3SM-SCM, WRF-LES, and ERA5 for the 28 February case.

Part of the reason for choosing the 1 March 2020 case is the well-represented CAO synoptic conditions (and forcings from reanalysis) and the good performance of MBL clouds in other models. As seen in the previous studies with WRF-CRM and WRF-LES models (Chen et al., 2022; Li et al., 2022, 2023), a single well-simulated case may still provide useful information on understanding the dynamics, thermodynamics, and aerosol-cloud-meteorology interactions over this region. We have also revised the structure of this paper and added a few more sensitivity tests and analyses to further understand the behavior of model physics in E3SM-SCM. On the other hand, we do have a plan of running long-term SCM simulations for statistical analysis over the ACTIVATE domain. However, including long-term statistical analysis in this CAO case study would defeat the purpose of evaluating the SCM against field observations and intercomparing SCM with CRM/LES results, so we decided to keep this paper as a single CAO case study and focus on the long-term analysis in a follow-up study.

**In the second half of the paper the authors state "…since Nd is underestimated it is difficult to demonstrate the value of adding aerosol vertical variation" which is blamed on weak vertical velocity updraft coming from the model. Why not do sensitivity experiments where the input vertical velocity to the aerosol activation is boasted by a certain factor to test the sensitivity?  This is the type of experiment that the SCM is ideal for.  In the conclusions the authors state "the evaluation of SCM simulations against the ACTIVATE measurements is helpful for understanding and improving turbulence representation over this region".  I don't think the authors have currently done this, but experiments that show the possible improvements/sensitivity with better turbulence linked to aerosol activation could provide justification for such a statement to be retained.**

Thank you for the great suggestion. We have now conducted a sensitivity test by directly enhancing the vertical velocity variance by a factor of 2, which makes the simulated turbulence close to the aircraft observation. As a result, the simulated cloud droplet number concentration and effective radius are closer to the aircraft observation (on top of the effect

of using observed aerosol number concentration). This result is now added to Section 4.1 and the plot is included as Fig. 8 (shown below).

[Figure]

Figure 8: (a) Vertical velocity variance <w'w'>, (b) cloud droplet number concentration $N_d$ and (c) cloud droplet effective radius $R_{eff}$ averaged between 15:00 and 16:00 UTC, when the aircraft measurements (shown in red crosses and boxes) were made. In the figure legend, "Climatology" is the original SCM simulation with prescribed aerosol concentration; "BCB2" is the SCM simulation with aerosol number concentration from the aircraft measurement at the BCB2 leg; and "2*<w'w'>" means the vertical velocity variance is enhanced by the factor of 2 in the SCM aerosol activation scheme.

**Section 4.3 ends with a statement that "E3SM SCM cannot provide information on sensitivity of aerosol to vertical distribution"… then why present results in this section at all? I feel adding a couple sentences or a paragraph to the conclusions/summary section stating that the authors attempted to address this in the SCM but couldn't because of x, y, and z would be good and sufficient… Which could help to motivate development of a validated aerosol model in the SCM.**

We have now removed this part from the main text, with only a brief statement in the Summary and Discussion section:

*"In the current SCM framework using observed aerosols, usually only one set of aerosol parameters, characterizing the spatially mean properties (i.e., particle number size distribution and composition), is fed into the model regardless of the aerosol vertical distribution (Liu et al., 2011; Liu et al., 2007; Klein et al., 2009; Lebassi-Habtezion and Caldwell, 2015; Li et al., 2023). The prescribed aerosol information based on observations is usually taken from in-situ measurements below the cloud base (e.g., Liu et al., 2011; Li et al., 2023), assuming that hygroscopic aerosol particles are readily activated into cloud droplets in the saturated air driven by updrafts. However, as aerosol concentration usually decreases with height in the lower atmosphere, regional aerosol vertical distribution may be changed*

*by in-cloud scavenging, horizontal transport, and vertical mixing, which can further affect cloud microphysical properties by secondary activation above cloud base (Wang et al., 2013; Wang et al., 2020). We conducted a sensitivity experiment with a specified aerosol vertical distribution (Fig. S5), but the configuration of prescribed aerosols in SCM only shows the response of clouds to aerosols given at the level of cloud formation. A more comprehensive consideration of complete aerosol processes (e.g., vertical transport, scavenging, deposition, etc.) is needed to include the cloud and dynamical feedback on aerosols and better understand the aerosol-cloud interactions."*